# HA stabilization promotes replication and transmission of swine H1N1 gamma influenza viruses in ferrets

Meng Hu[1], Guohua Yang[1], Jennifer DeBeauchamp[1], Jeri Carol Crumpton[1], Hyunsuh Kim[1], Lei Li[2†], Xiu-Feng Wan[2,3,4,5,6,7], Lisa Kercher[1], Andrew S Bowman[8], Robert G Webster[1], Richard J Webby[1,9], Charles J Russell[1,9]*

[1]Department of Infectious Diseases, St. Jude Children's Research Hospital, Memphis, United States; [2]Department of Basic Sciences, College of Veterinary Medicine, Mississippi State University, Mississippi State, United States; [3]Missouri University Center for Research on Influenza Systems Biology (CRISB), University of Missouri, Columbia, United States; [4]Department of Molecular Microbiology and Immunology, School of Medicine, University of Missouri, Columbia, United States; [5]Bond Life Sciences Center, University of Missouri, Columbia, United States; [6]Department of Electrical Engineering Computer Science, College of Engineering, University of Missouri, Columbia, United States; [7]MU Informatics Institute, University of Missouri, Columbia, United States; [8]Department of Veterinary Preventive Medicine, The Ohio State University, Columbus, United States; [9]Department of Microbiology, Immunology & Biochemistry, College of Medicine, The University of Tennessee Health Science Center, Memphis, United States

*For correspondence:
charles.russell@stjude.org

Present address: †Department of Medicine, Section of Rheumatology, the Knapp Center for Lupus and Immunology, University of Chicago, Chicago, United States

**Competing interests:** The authors declare that no competing interests exist.

**Abstract** Pandemic influenza A viruses can emerge from swine, an intermediate host that supports adaptation of human-preferred receptor-binding specificity by the hemagglutinin (HA) surface antigen. Other HA traits necessary for pandemic potential are poorly understood. For swine influenza viruses isolated in 2009–2016, gamma-clade viruses had less stable HA proteins (activation pH 5.5–5.9) than pandemic clade (pH 5.0–5.5). Gamma-clade viruses replicated to higher levels in mammalian cells than pandemic clade. In ferrets, a model for human adaptation, a relatively stable HA protein (pH 5.5–5.6) was necessary for efficient replication and airborne transmission. The overall airborne transmission frequency in ferrets for four isolates tested was 42%, and isolate G15 airborne transmitted 100% after selection of a variant with a stabilized HA. The results suggest swine influenza viruses containing both a stabilized HA and alpha-2,6 receptor binding in tandem pose greater pandemic risk. Increasing evidence supports adding HA stability to pre-pandemic risk assessment algorithms.

## Introduction

Aside from recent bat isolates (*Tong et al., 2012*; *Tong et al., 2013*; *Kandeil et al., 2019*), influenza A viruses (IAVs) originate from avian species and transmit to humans directly or via other animals (e.g. swine) (*Long et al., 2019b*; *Russell et al., 2018*). Genetically and antigenically diverse IAV strains circulate in wild and domestic birds, aquatic mammals, canine, swine, and humans (*Long et al., 2019b*). IAVs contain eight RNA gene segments (*Shaw and Palese, 2013*; *Noda et al., 2018*), and three previous human pandemic viruses (1957, 1968, and 2009) underwent gene reassortment before spreading to humans (*Smith et al., 2009*; *Garten et al., 2009*). Swine can serve as an intermediate host between avian species and humans in part by expressing both avian IAV-

preferred (α2,3-linked) and human IAV-preferred (α2,6-linked) sialic-acid-terminated receptors (*Rajao et al., 2018*; *de Graaf and Fouchier, 2014*). Currently, H1N1, H1N2, and H3N2 subtypes are endemic in swine (*Gao et al., 2017*; *Walia et al., 2019*), and it is challenging to triage these genetically diverse viruses for pre-pandemic countermeasures.

In addition to pandemics, swine IAVs cause sporadic spillovers from swine to humans. More than 400 cases of swine-IAV infections in humans have been documented between 2005 and 2018 (*World Health Organization (WHO), 2018*). H1N1-subtype IAVs are currently enzootic in swine, constituting a pool of genetically diverse and continuously evolving strains. Swine H1N1 strains are divided into three major lineages: classical swine (1A), human seasonal (1B), and Eurasian avian (1C) (*Anderson et al., 2016*). The classical swine lineage branches into three clades based on early regional considerations: alpha (α; 1A.1), beta (β; 1A.2), and gamma (γ; 1A.3) (*Anderson et al., 2016*). In approximately 1999, gamma viruses split into two branches: swine gamma (1A.3.3.3) and swine viruses that later contributed the hemagglutinin (HA) gene to the 2009 human pandemic virus (1A.3.3.2). (*Gao et al., 2017*). Frequently after 2009, HA and other gene segments from H1N1pdm viruses transmitted from humans to swine, generating diverse reassortant viruses. Swine gamma viruses isolated after 2009 contain varying numbers of internal genes originating from human H1N1pdm viruses (*Ducatez et al., 2011*). Here, we studied H1N1pdm and gamma clade viruses because these two clades contain HA genes that are closely related to the 2009 pandemic and many gamma clade viruses have one or more H1N1pdm internal genes that are already human adapted.

During IAV entry, HA trimers bind to sialic-acid-terminating moieties, triggering endocytosis of virions. Avian IAVs have HA proteins evolved to bind glycoprotein and glycolipid receptors having terminal sialic acids with an $\alpha-2,3$ linkage, and human-adapted IAVs have HA proteins containing receptor-binding pocket mutations that allow higher affinity binding to $\alpha-2,6$-linked sialic acid (*de Graaf and Fouchier, 2014*; *Xiong et al., 2014*; *Gambaryan and Matrosovich, 2015*). Within minutes after endocytosis, virions are exposed to increasing acidification (*Mellman et al., 1986*; *Cain et al., 1989*). Upon reaching a threshold pH that varies by virus and typically ranges from pH 4.8 to 6.2 (*Galloway et al., 2013*; *Russell et al., 2018*), spring-loaded HA nanomachines are triggered to irreversibly change their structures and fuse the viral envelope with the endosomal membrane (*Skehel and Wiley, 2000*). Thus, IAV entry is regulated by HA receptor binding specificity and HA stability.

Avian IAVs typically have HA proteins triggered at pH 5.5–6.2 while H1N1pdm isolates (2010–2011) have been shown to trend lower at pH 5.3–5.5 (*Russell et al., 2018*). This suggests that avian-to-human adaptation of the HA protein may include a decrease in activation pH. Multiple mechanisms may be operational. HA proteins with a lower activation pH have increased resistance to extracellular inactivation and have increased environmental stability (*Poulson et al., 2016*; *Labadie et al., 2018*; *Russier et al., 2020*). This may be important because the human upper respiratory tract is mildly acidic (*England et al., 1999*; *Man et al., 2017*). Intracellularly, opposing intracellular forces may select for intermediate HA stability (*Singanayagam et al., 2019*), which may help minimize interferon responses both in dendritic cells triggered by high-pH HA proteins (*Russier et al., 2020*) and in late endosomes triggered by low-pH HA proteins via interactions with IFITM2 and IFITM3 proteins (*Gerlach et al., 2017*). Consequently, an optimal HA stability is essential for virus replication.

A pandemic-capable IAV requires its HA protein has been unseen immunologically by the majority of the population and is able to support efficient replication and transmission between humans. Pandemic risk assessment tools developed by the Centers for Disease Control and Prevention (CDC) and World Health Organization (WHO) consider virologic, population, and ecological/epidemiological factors (*Centers for Disease Control and Prevention (CDC), 2019* and *World Health Organization (WHO), 2016*). These are called IRAT and TIPRA, respectively. The algorithms explicitly consider HA receptor-binding specificity; however, other virologic properties that may be important are not included or are only considered indirectly by virtue of transmission experiments (reviewed in *Russell et al., 2018*; *Lipsitch et al., 2016*; *Long et al., 2019b*). Presently, HA stability is not considered in risk assessment tools but has been linked to IAV transmission and host adaptation. HA stabilization occurred after H1N1pdm circulation in humans (*Maurer-Stroh et al., 2010*; *Russier et al., 2016*) and has been shown to be necessary for airborne transmission of avian H5N1 and 2009 H1N1 pandemic (H1N1pdm) viruses in ferrets (*Russier et al., 2016*; *Imai et al., 2012*; *Herfst et al., 2012*). As swine provide an established pathway for the emergence of pandemic influenza and HA stability

is an intrinsic property of the key gene involved in pandemics, we considered it important to determine the impact of HA stability on swine IAV replication and transmissibility.

In this study, we hypothesized that contemporary swine viruses vary in HA stability and strains having more stable HA proteins are more likely to transmit by the airborne route in ferrets. To test this hypothesis, we measured HA activation pH for a panel of H1N1 swine gamma viruses and infected ferrets with genetically related pairs of H1N1 swine gamma viruses that differed in HA stability.

## Results

### H1N1 gamma isolates from swine had higher HA activation pH values than pandemic isolates from swine and humans

To survey HA stability of contemporary H1N1 influenza A viruses (IAVs), we collected 34 swine and 21 human H1N1 IAVs (2009–2016) (*Supplementary file 1*). Phylogenetic analyses of HA segments showed that 22 swine isolates were gamma clade while 21 human and 12 swine isolates were pandemic (pdm) clade (*Figure 1—figure supplement 1*). All isolates were widely distributed within gamma and pdm clades and thus could be taken as genetically diverse representatives of the two targeted genetic clades (*Figure 1—figure supplement 1*).

We measured HA stability by syncytia assay, recording the highest pH at which virus-expressed HA proteins were induced to cause membrane fusion. A/Tennessee/1-560/2009 (H1N1) (A/TN/2009) had an HA activation pH of 5.5, a moderately stable value previously reported for human pandemic isolates at the start of the 2009 pandemic (*Russier et al., 2016*; *Galloway et al., 2013*). The pdm clade viruses isolated from humans and swine after 2009 had HA activation pH values ranging from 5.0 to 5.5 with an average of 5.3 (*Figure 1A*, *Supplementary file 1*). In contrast, gamma clade swine viruses had significantly higher HA activation pH values that ranged from 5.5 to 5.9 and averaged 5.7 (*Figure 1A*, p<0.0001 compared to the pdm clade). Overall, pdm-origin HA proteins were more stable than swine gamma, and HA stability for the two clades overlapped at pH 5.5.

### Swine gamma viruses had higher average peak titers in MDCK cells

To evaluate replication capacities of the viruses in vitro, we inoculated viruses into MDCK cells at a multiplicity of infection (MOI) of 0.01 PFU/cell and measured virus titers at 12, 24, and 36 hr post infection (hpi). A/TN/2009, a reference isolate, had a peak titer of $3.63 \times 10^6$ TCID$_{50}$/ml (*Figure 1B*, open square). Peak titers of human pdm viruses ($2.86 \times 10^5$ to $3.29 \times 10^7$ TCID$_{50}$/ml) were similar to those from swine pdm isolates ($7.87 \times 10^6$ to $6.76 \times 10^7$ TCID$_{50}$/ml; p=0.937). However, peak titers of swine gamma viruses ($2.80 \times 10^6$ to $6.56 \times 10^8$ TCID$_{50}$/ml) averaged 10-fold higher than pdm clade (p<0.001, *Figure 1B*, *Figure 1—figure supplement 2*).

### Two similar gamma virus pairs differed in HA stability

For further work, we selected two pairs of gamma viruses that within each pair contained (a) substantial differences in HA stability, (b) only one amino-acid substitution in HA, and (c) limited amino-acid variations in other genes. Pair 1 consisted of A/swine/Illinois/2A-1213-G15/2013 (G15) and A/swine/Illinois/2B-0314-P4/2014 (P4), which contain NP and M genes from the pdm clade and the six other genes (including HA) from swine gamma (*Figure 2A*). Consensus sequences of these viruses differed at amino-acid positions HA1-210, PA-271, and PB2-648. HA1-210 is located at the HA receptor-binding domain near the trimer interface, where it could potentially modulate HA head dissociation during acid-induced activation (*Figure 2B*). Next-generation sequencing (completed after ferret studies) revealed isolate P4 contained approximately 100% HA1-S210 while isolate G15 contained approximately 85% HA1-N210 and 15% HA1-S210 (*Figure 2C*). Virus isolates P4 and G15 induced syncytia formation at pH 5.5 and 5.8, respectively, whether the HA and NA genes were expressed from the background of wild-type viruses or 6+2 reassortant viruses containing internal genes from A/TN/2009 (*Figure 3A*). Thus, the PA and PB2 variations did not appear to contribute to differences in HA stability. Acid inactivation titrations showed G15 virus was susceptible to inactivation at 0.2 pH units higher than P4 (*Figure 3B*). Viruses G15 and P4 grew to similar titers that were nearly 100-fold higher than A/TN/2009 in swine testicular (ST) and MDCK cells (*Figure 3C*). The Pair 1 viruses bound to both α2,6- and α2,3-linked receptors (*Figure 3D*). P4 had higher binding to α2,6 and less binding

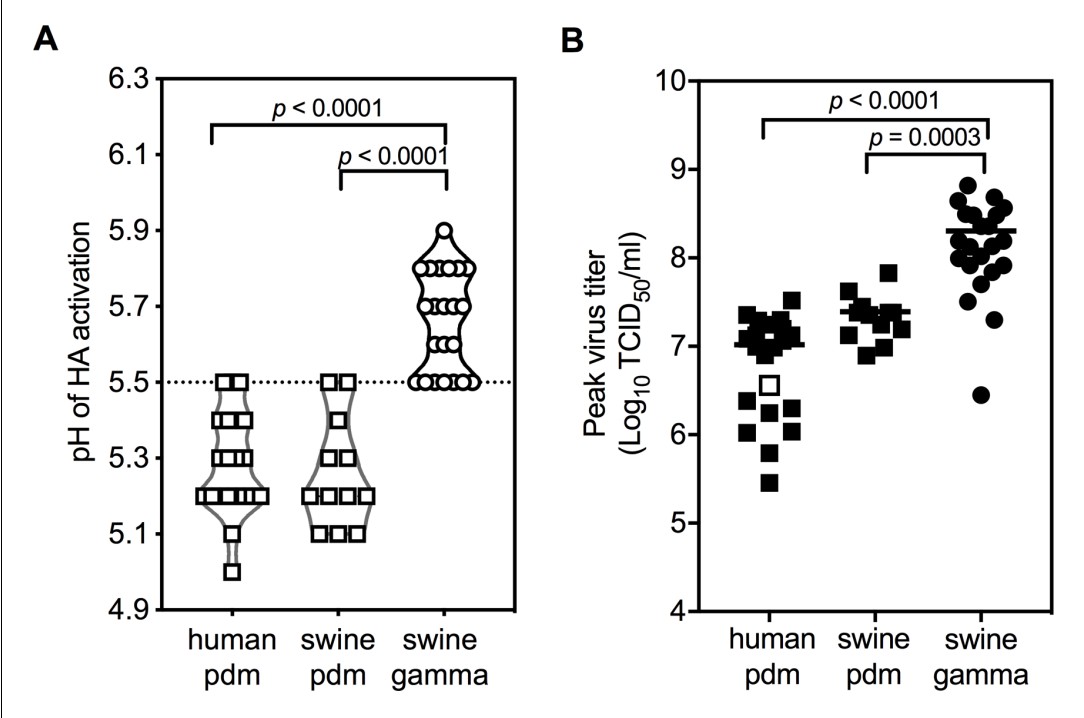

**Figure 1.** HA activation pH values and replication capacities of H1N1 pandemic (pdm) and gamma virus isolates from humans and swine. (**A**) HA activation pH values. Vero cells were infected with IAVs (human pdm n = 21; swine pdm n = 12; swine gamma n = 22) at a multiplicity of infection (MOI) of 3 PFU/cell. At 16 hr post-infection, HA activation pH values were measured by syncytia assay. Each symbol represents the mean HA activation pH value of an individual isolate. A dashed line is shown at pH 5.5. (**B**) Replication capacities in MDCK cells. MDCK cells were infected by pandemic and gamma viruses at an MOI of 0.01 PFU/cell. Cell culture supernatants were harvested at 12, 24, and 36 hr, and were titrated by $TCID_{50}$ on MDCK cells. Each symbol represents the mean peak titer of an individual isolate. The white square in the human pdm group is the reference virus A/TN/09. All experiments were independently performed at least twice. *P* values were determined according to one-way ANOVA followed by a Tukey's multiple comparisons test.

The online version of this article includes the following figure supplement(s) for figure 1:

**Figure supplement 1.** Phylogenic analyses of HA segments of swine and human H1N1 influenza A viruses (2009–2016).
**Figure supplement 2.** Virus replication in MDCK cells.

to α2,3 than G15. In summary, Pair 1 gamma swine viruses were capable of efficient replication in cultured cells and binding to both α2,6- and α2,3-linked receptors. However, P4 had an HA activation pH of 5.5 and receptor-binding skewed toward an α2,6 preference, whereas G15 had a substantially higher activation pH of 5.8 and receptor-binding skewed toward an α2,3 preference.

Pair 2 consisted of A/swine/Illinois/2E-0113-P19/2013 (P19) and A/swine/Illinois/2E-0113-P24/2013 (P24), which contain PB2, PB1, PA, and M genes from the pdm clade and the four other genes (including HA) from swine gamma (*Figure 2A*). Consensus sequences of these viruses had only one major amino-acid variation in the genome, residue HA2-117. HA2-117 is located in helix D of the central triple-stranded coiled coil in the stalk, where it forms part of a pocket engaged by the fusion peptide of an adjacent protomer in the prefusion structure (*Figure 2B*). Next-generation sequencing (completed after ferret studies) showed that isolate P24 contained approximately 100% HA2-N117 while isolate P19 contained approximately 83% HA2-T117 and 17% HA2-N117 (*Figure 2D*). Virus isolates P24 and P19 caused syncytia formation at pH 5.6 and 5.9, respectively (*Figure 4A*). In the background of 6 + 2 reassortment viruses containing internal genes from A/TN/2009, P24 caused membrane fusion at pH 5.6 while P19 was shifted up to 6.0 (*Figure 4A*). P19 virus was susceptible to acid inactivation at 0.15 pH units higher than P24 (*Figure 4B*). Viruses P24 and P19 grew to similar titers and approximately 10 to 100-fold higher than A/TN/2009 in ST and MDCK cells (*Figure 4C*). The Pair 2 viruses bound to both α2,6- and α2,3-linked receptors similarly (*Figure 4D*). In summary, the Pair 2 viruses replicated efficiently in cell culture and bound similarly to α2,6- and α2,3-linked

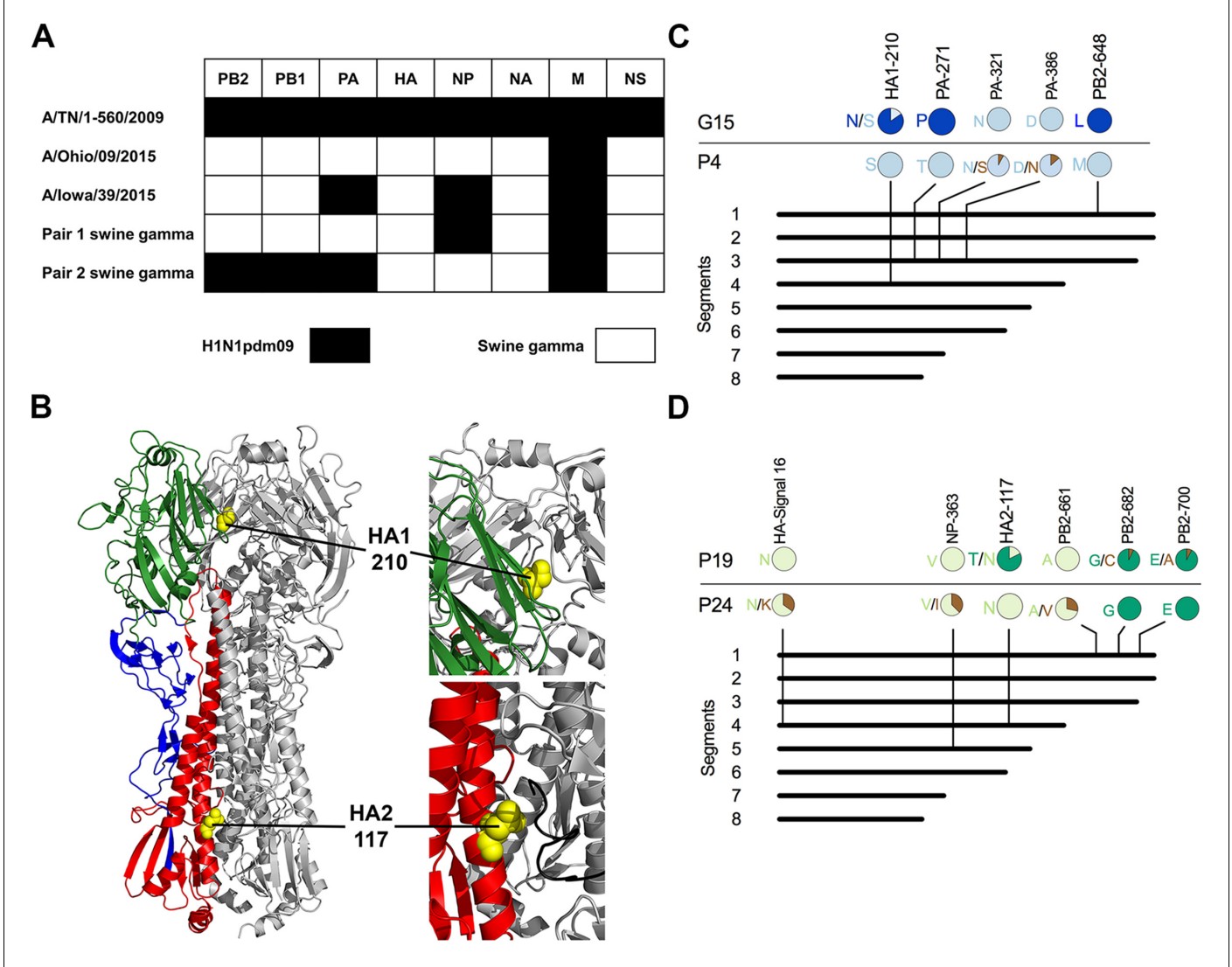

**Figure 2.** Genetic, structural, and sequence analyses of swine gamma Pair 1 and Pair 2 viruses. (**A**) Lineages of Pair 1 and 2 viruses. Virus gene segments were analyzed by using CLC Genomics Workbench version 11.0.1. A/Ohio/09/2015 and A/Iowa/39/2015 H1N1 viruses are gamma viruses from human infections (*Pulit-Penaloza et al., 2018*). Gene segments from pandemic and swine endemic are marked in black and white, respectively. (**B**) HA protein structure and locations of HA residue variations. HA1 residue 210 (yellow) is located in the HA receptor-binding domain head (green) in contact with an adjacent protomer (gray). HA2 residue 117 (yellow) is located in the stalk region (red) in contact with the fusion peptide (black) of an adjacent protomer. HA1 residues in the stalk are colored blue, and two protomers of the trimer are colored gray. The structure was generated by Mac PyMOL using A/California/04/2009 (H1N1) (PDB entry 3UBE). (**C**) Sequence variations between Pair 1 isolates G15 and P4. (**D**) Sequence variations between Pair 2 isolates P19 and P24. Whole-genome analyses were performed by next-generation sequencing. Amino-acid variations with frequencies greater than 5% are shown in colored pie charts. H3 numbering was used for the HA segment.

receptors. However, the Pair 2 viruses differed at stalk residue HA2-117, which resulted in P24 becoming activated at pH 5.6 for membrane fusion while P19 was activated at pH 5.9.

## Pair 1 viruses showed consistently efficient transmissibility by contact but varying transmissibility through airborne exposures in ferrets

To investigate the impact of HA stability on virus pathogenicity and transmission, three donor ferrets were intranasally inoculated with $10^6$ PFU of isolates G15 and P4, which had a low HA stability (activation pH 5.8) and a moderate HA stability (activation pH 5.5), respectively. Each group was in an isolated cubicle, and the G15 cubicle was processed before decontamination and handling the P4 group. Additionally, each group had three cages containing one donor ferret each. One day

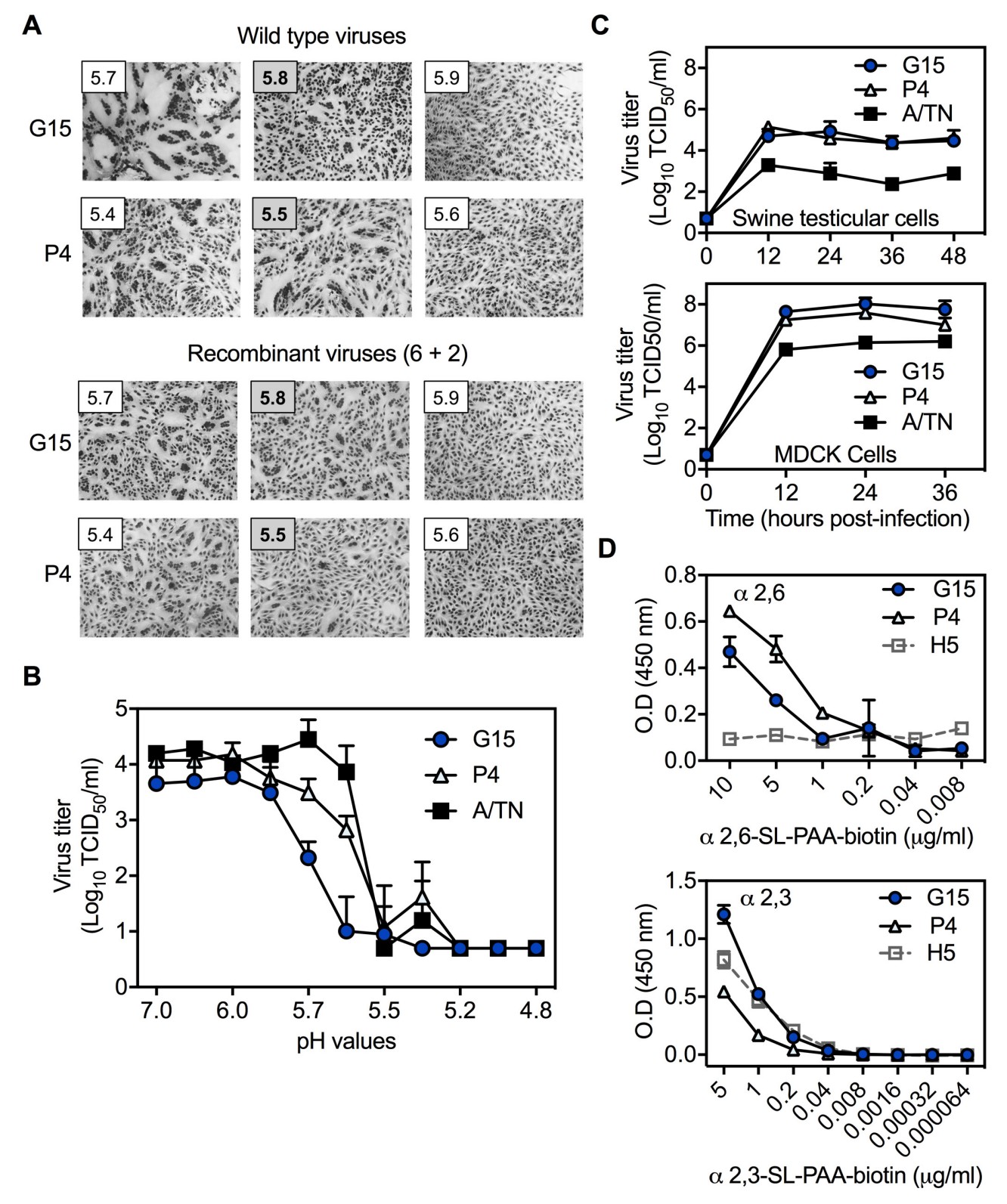

**Figure 3.** Pair 1 (G15 and P4) virus characterization in vitro. (**A**) HA activation pH measured by syncytia assay. Viruses were inoculated into Vero cells at an MOI of 3 PFU/cell. Recombinant viruses with HA and NA segments from G15 and P4 were rescued in the background of the six internal genes from A/TN/09 using reverse genetics. Representative images of three independent experiment results are shown. (**B**) HA inactivation pH measured by loss of infectivity as a function of acid exposure. Virus aliquots were treated with pH-adjusted PBS, re-neutralized, and subjected to measurement of residual

*Figure 3 continued on next page*

Figure 3 continued

virus infectivity by TCID$_{50}$. (C) Virus growth capacity in swine testicular (ST) and MDCK cells. Viruses were inoculated into ST and MDCK cells at an MOI of 0.01 PFU/cell. Cell culture supernatants were harvested at the indicated time points and quantified by TCID$_{50}$. (D) Binding specificities toward α(2, 6)- or α(2, 3)-linked sialic acid receptors. Viruses were inoculated onto fetuin-coated plates. Virus binding affinities toward α2,6- or α2,3-linked sialylglycopolymers were measured by solid-phase receptor binding assay. Recombinant A/Puerto Rico/9/1934 (H1N1) with the HA segment from A/Mallard/Alberta/383/2009 (H5N1) was used as a control. All data were means ± SD of at least two independent experiments.

following inoculation, three naive contact ferrets were cohoused with the donor ferrets, and three airborne ferrets were introduced individually into adjacent non-contact compartments separated by perforated dividers that allowed air flow. All ferrets were monitored for body weight and temperature daily, and nasal washes were collected every other day until day 14. Ferret sera were collected on day 21. The results showed that temperature and body weight changes of ferrets infected with G15 and P4 were limited and similar to each other (*Figure 5—figure supplement 1*), consistent with swine viruses typically causing only mild disease in ferrets (*Russier et al., 2017*; *Pulit-Penaloza et al., 2018*). On day 1, ferrets inoculated with G15 had nasal wash titer of $1.5 \times 10^5$ TCID$_{50}$/ml, whereas those from P4-infected ferrets were 10-fold higher ($2.2 \times 10^6$ TCID$_{50}$/ml, p=0.23) (*Figure 5A*). Peak days (day 2.3 and 3.0 for G15 and P4, respectively, p=0.21) and peak titers of infection (6.4 and 6.7 log$_{10}$ TCID$_{50}$/ml for G15 and P4, respectively) were similar (*Table 1*). Moreover, both groups displayed similar contact transmission efficiency (3/3), contact virus growth, and contact peak nasal titers (6.95 and 7.90 log$_{10}$ TCID$_{50}$/ml for G15 and P4, respectively, p=0.40) (*Figure 5B*). With respect to airborne transmission, only 1/3 ferrets in the P4 group (activation pH 5.5) seroconverted and had detectable nasal wash virus, which peaked at $2.32 \times 10^5$ TCID$_{50}$/ml (*Figure 5C–D*). The G15 (pH 5.8) airborne group had 3/3 seroconversion and 2/3 nasal virus isolation (peak titers of $2.32 \times 10^6$ TCID$_{50}$/ml and 50 TCID$_{50}$/ml). In summary, both virus isolates replicated efficiently in donors, transmitted efficiently to contacts, and replicated to high titers in contacts. Ferrets inoculated with G15 supported efficient airborne transmission (3/3) while those inoculated with P4 had only partial (1/3).

## Stabilizing variant HA1-S210 outgrew in the G15-infected and transmitted ferrets

Increased airborne transmission by the G15 group was unexpected, as G15's predominant HA1-N210 population resulted in a higher HA activation pH (5.8) and a decreased binding ratio of α2,6- versus α2,3-linked receptors compared to P4. Using viruses collected from ferret nasal washes, we measured HA stability by syncytia assay. For the group inoculated with P4 isolate (pH 5.5), HA activation pH values in donor, contact, and airborne animals remained at an intermediate stability of pH 5.5–5.6 (*Figure 6A*). The HA activation pH of the G15 inoculum was 5.8. G15-infected donor and exposed contact ferrets yielded nasal viruses with HA activation pH ranging from 5.6 to 5.9. While three G15-exposed airborne ferrets seroconverted and viruses were recovered from two, only one airborne ferret yielded sufficient virus for phenotypic analysis, which revealed a value of 5.6 on the first day collected (day 8, *Figure 6A*). Thus, P4 (starting with a moderately stable HA) retained its phenotype after transmission while G15 (starting with an unstable HA) had its HA protein become stabilized during airborne transmission.

We sequenced ferret nasal washes by next-generation sequencing. The percentages of HA variants (synonymous and nonsynonymous) occurring in G15 donor and contact ferrets were significantly higher than those of P4 (*Figure 6B*), showing G15 was less genetically stable in ferrets. P4 had several nonsynonymous mutations arise in donor, contact, and airborne ferrets; however, none of these variants increased over time or became dominant (*Figure 6—figure supplement 1A–C*). G15 also had a variety of minor variants that did not become dominant or increase substantially from day-to-day (*Figure 6—figure supplement 1D–F*). As described in *Figure 2C*, the inoculum of G15 contained 85% HA1-N210 and 15% HA1-S210. HA1-S210 outgrew HA1-N210 within one day in two inoculated ferrets and by 3 days in the remaining ferret (*Figure 6C*, *Figure 6—figure supplement 1D*). In contact and airborne G15 ferrets, >99% variants consisted of HA1-S210. Contamination by P4 (which also contained HA1-S210) was ruled out, as HA1-S210-containing G15-group samples retained identifying PA-P271 and PB2-L648 residues that were not in P4. Collectively, airborne transmission for both Pair 1 groups occurred with a frequency of 4/6 seroconversion (3/6 virus isolation),

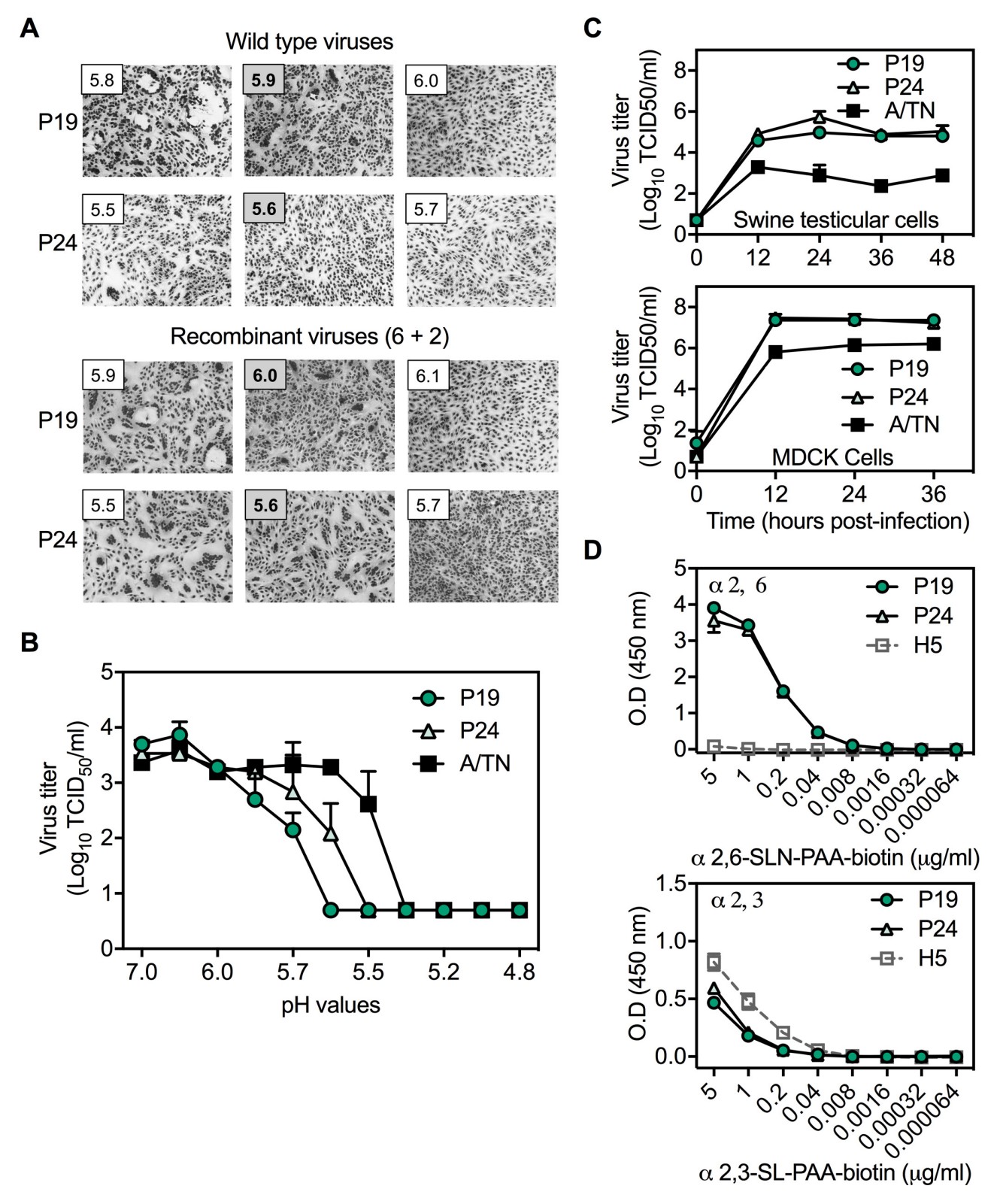

**Figure 4.** Pair 2 (P19 and P24) virus characterization in vitro. (A) Syncytia assay results for wild-type and 6+2 reassortant viruses. (B) HA inactivation pH values quantified by $TCID_{50}$. (C) Virus replication in ST and MDCK cells inoculated at an MOI of 0.01 PFU/cell and quantified by $TCID_{50}$. (D) Receptor binding specificities to α2,6- or α2, 3-linked sialic acid receptors. All data were mean ± SD of at least two independent experiments.

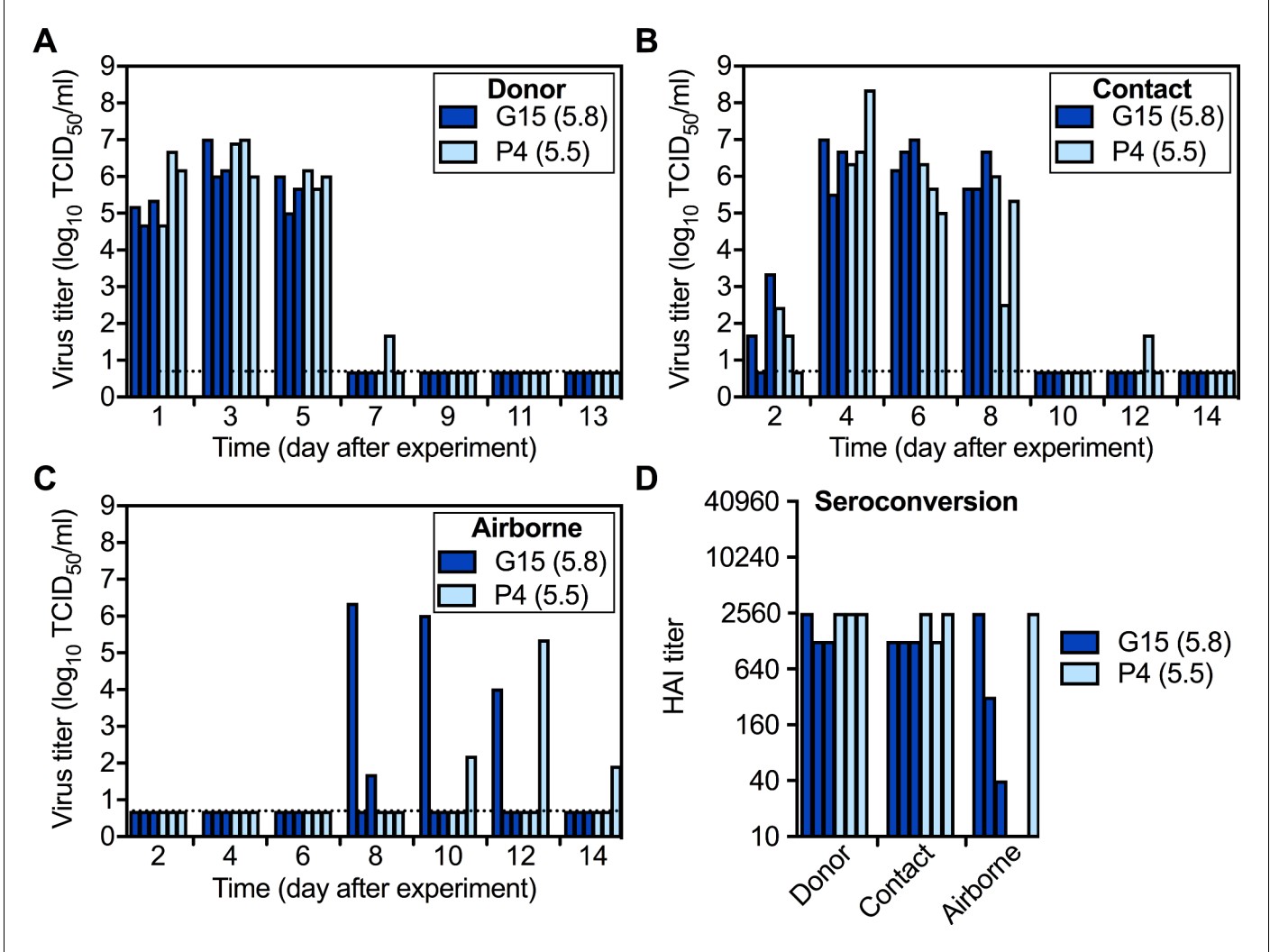

**Figure 5.** Pair 1 (G15 and P4) replication and transmission in ferrets. Three inoculated (Donor) ferrets were caged separately. On the next day, 3 naive Contact ferrets each were cohoused with the Donor and 3 naive Airborne ferrets were placed into perforated cages allowing transmission through the air. (A–C) Virus $TCID_{50}$ titers of nasal washes from Donor (A), Contact (B), and (C) Airborne ferrets. Each bar represents the virus titer of an individual sample, and the dashed lines represent the limit of detection. (D) HAI titers of day-21 sera of ferrets infected or exposed to the tested viruses. The difference of 3/3 versus 1/3 airborne-transmission events was not statistically significant (p>0.05).

The online version of this article includes the following figure supplement(s) for figure 5:

**Figure supplement 1.** Clinical symptoms in ferrets upon infection with Pair 1 viruses.

transmitted virus contained HA1-S210, and the HA activation pH of airborne-transmitted virus was 5.5–5.6 upon first isolation. In the G15 group, virus containing the unstable HA1-N210 variation (pH 5.8) was outcompeted and did not airborne transmit in ferrets; instead the minor HA1-S210 (pH 5.5) sub-population was airborne transmitted along with its unique PA-P271 and PB2-L648 variations.

## HA stabilization was also favored during infection and transmission with Pair 2 viruses

Compared to HA1-N210, the HA1-S210 subpopulation that rapidly became dominant in G15-inoculated ferrets had increased HA stability and increased binding to α2,6- versus α2,3-linked receptors (*Figure 3D*). Both these phenotypic changes have been shown to promote IAV replication and transmission in ferrets (*Russell et al., 2018*). Therefore, we performed a ferret transmission study using Pair 2 isolates P19 and P24 that had only one nonsynonymous variation at HA2-117 that modulated

**Table 1.** H1N1 swine gamma virus characterization before and after infection and transmission in ferrets.

| Virus characterization | | Pair 1 viruses | | Pair 2 viruses | |
| --- | --- | --- | --- | --- | --- |
| | | G15 | P4 | P19 | P24 |
| HA activation pH[*] | Wild-type viruses | 5.8 | 5.5 | 5.9 | 5.6 |
| | Recombinant (6+2)[†] | 5.8 | 5.5 | 6.0 | 5.6 |
| HA inactivation pH[‡] | Wild-type viruses | 5.8 | 5.6 | 5.9 | 5.7 |
| Genome variations | | HA1-N210 PB2-L648 PA-P271 | HA1-S210 PB2-M648 PA-T271 | HA2-T117 | HA2-N117 |
| Replication in vitro | ST cells | Similar | | Similar | |
| | MDCK cells | Similar | | Similar | |
| Receptor binding specificity[§] | α2, 6 | Lower | Higher | Similar | |
| | α2, 3 | Higher | Lower | Similar | |
| Donor ferrets | Viruses isolated | 3/3 | 3/3 | 3/3 | 3/3 |
| | Seroconversion | 3/3 | 3/3 | 3/3 | 3/3 |
| | Day of peak titer | 3 | 2.3 (±1.2) | 4.3 (±1.2) | 1.7 (±1.2) |
| | $P$ (t-test), peak titer day | 0.21 | | 0.047 | |
| | Peak titers ($\log_{10}$ TCID$_{50}$) | 6.4 (±0.54) | 6.7 (±0.45) | 6.6 (±0.19) | 6.9 (±0.89) |
| | HA activation pH range | 5.6–5.9 | 5.5–5.6 | 5.6–6.0 | 5.5-.6 |
| | Major mutants | HA1-N210S | None | HA2-T117N | None |
| Contact ferrets | Viruses isolated | 3/3 | 3/3 | 3/3 | 3/3 |
| | Seroconversion | 3/3 | 3/3 | 3/3 | 3/3 |
| | HA activation pH range | 5.6–5.9 | 5.5–5.6 | 5.6–5.9 | 5.6–5.7 |
| | Major mutants | HA1-N210S | None | HA2-T117N | None |
| Airborne ferrets | Viruses detected | 2/3 | 1/3 | 1/3 | 0/3 |
| | Seroconversion | 3/3 | 1/3 | 1/3 | 0/3 |
| | HA activation pH[¶] | 5.6 | 5.5 | 5.6 | NA[**] |
| | Major variants | HA1-N210S | None | HA2-T117N | NA |

[*]HA activation pH measured by syncytia assay.

[†]Recombinant 6+2 viruses contained HA and NA genes from swine gamma isolates and the six internal genes from A/TN/1-560/09 (H1N1).

[‡]HA inactivation pH measured by acid-induced inactivation pH with TCID$_{50}$ readout.

[§]Receptor binding specificity measured by solid-phase receptor binding assay.

[¶]HA activation pH of airborne-transmitted virus on first day of isolation.

[**]NA, Not applicable.

HA stability but not receptor-binding specificity (*Figure 4*). The results for Pair 2 largely resembled those for Pair 1, albeit with a lower frequency of airborne transmission.

After one day of infection in inoculated donors, the average nasal wash titer of P24 was 250-times higher than that of P19 ($3.0 \times 10^7$ TCID$_{50}$/ml and $1.2 \times 10^5$ TCID$_{50}$/ml, respectively) (*Figure 7A*). This difference was not statistically significant (p=0.33), and both groups had similar titers on day 3. Peak titers were attained approximately 2.5 days quicker in the P24 group (p=0.047) but were similar in magnitude for both groups (6.6 and 6.9 $\log_{10}$ TCID$_{50}$ for P19 and P24, respectively, *Table 1*). P24 donor ferrets had slightly higher body temperatures on several days and small increases in weight loss compared to P24 (*Figure 7—figure supplement 1*), but these differences were relatively small and both groups had only mild symptoms of infection. Both P19 and P24 transmitted with 100% efficiency by contact (*Figure 7B,D*), and virus titers were comparable between groups (*Figure 7B*). Airborne transmission was less efficient for these viruses than for Pair 1, as Pair 2 virus isolation and seroconversion were detected for only 1/3 ferrets in the P19 group and 0/3 in the P24 group (*Figure 7C,D*).

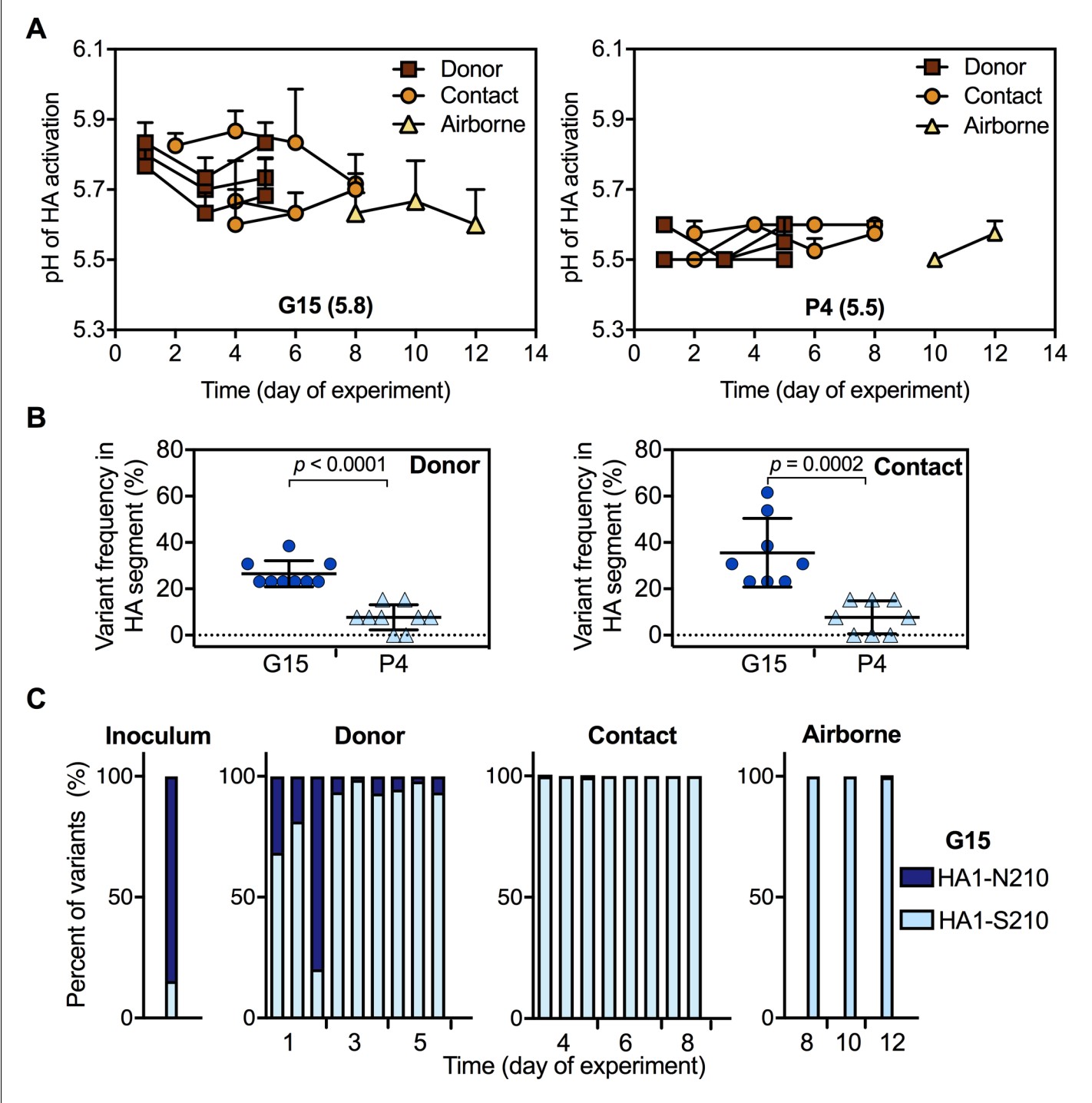

**Figure 6.** Phenotypes and genotypes of Pair 1 (G15 and P4) viruses after infection and transmission in ferrets. The transmission study was performed as described in *Figure 5*. (A) Virus HA activation pH values (means ± SD) measured by at least two independent syncytia assay experiments. (B) HA variant percentages in G15 and P4 viruses after infection and contact transmission in ferrets calculated by using mutations (synonymous and nonsynonymous) with a frequency larger than 5% during the transmission study in comparison to the consensus sequences of the initial inoculums. *P* values were determined by Mann-Whitney U test. (C) Proportions of HA1-N210 and HA1-S210 in G15 before and after infection and transmission in ferrets. The online version of this article includes the following figure supplement(s) for figure 6:

**Figure supplement 1.** Heat maps of mutations detected in nasal washes of Pair 1 ferrets.

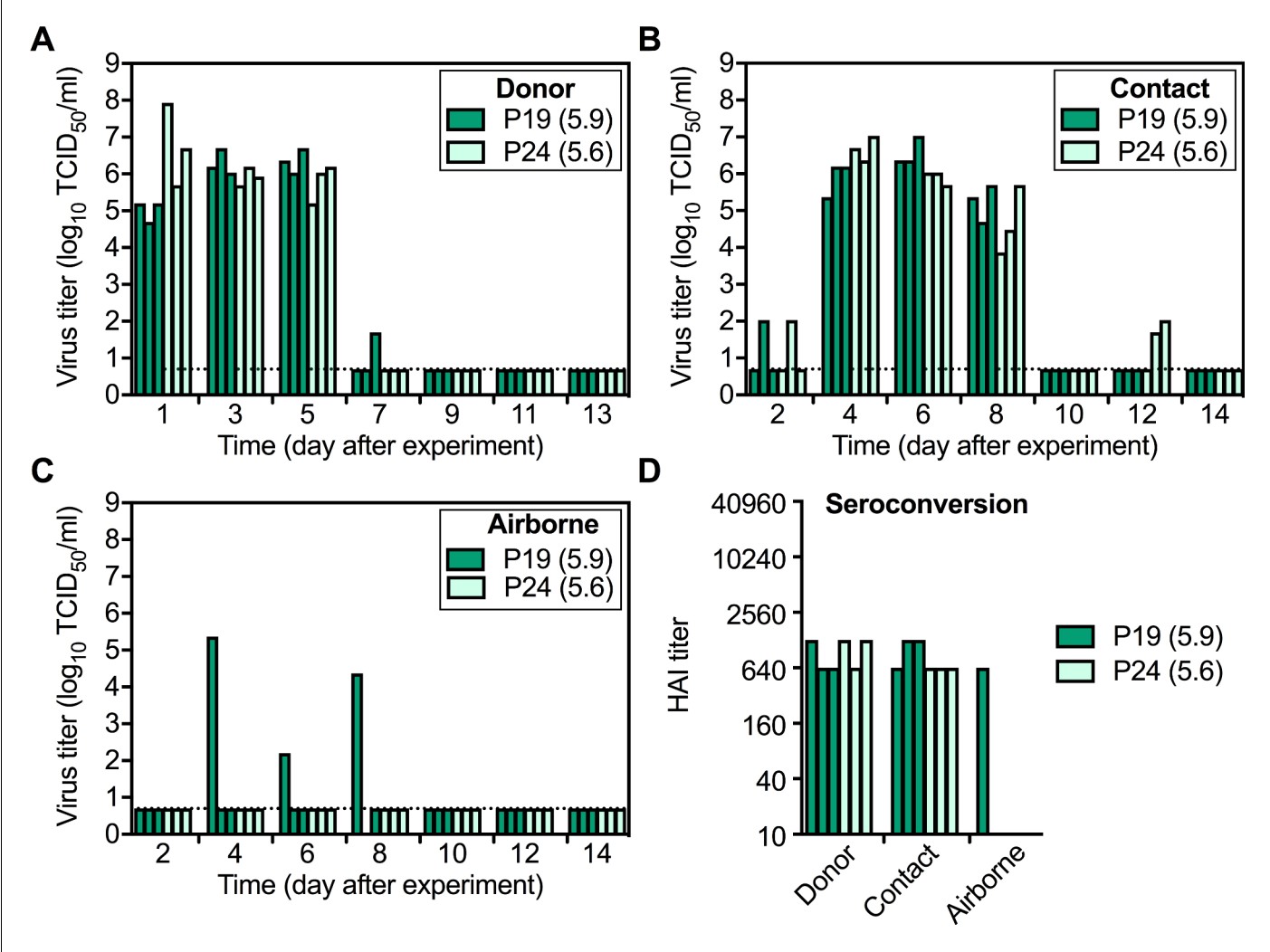

**Figure 7.** Pair 2 (P19 and P24) replication and transmission in ferrets. The transmission study was performed as described in *Figure 5*. (A–C) Virus TCID$_{50}$ titers of nasal washes from Donor (A), Contact (B), and (C) Airborne ferrets. Each bar represents the virus titer of an individual sample, and the dashed lines represent the limit of detection. (D) HAI titers of day-21 sera of ferrets infected or exposed to the tested viruses. The difference of 1/3 versus 0/3 airborne-transmission events was not statistically significant (p>0.05).

The online version of this article includes the following figure supplement(s) for figure 7:

**Figure supplement 1.** Clinical symptoms in ferrets upon infection with Pair 2 viruses.

HA activation pH of viruses recovered from donor and contact ferrets in the P24 group were maintained at pH 5.6–5.7 (*Figure 8A*). In the P19 group inoculated with virus having an HA activation pH of 5.9, HA stability in one donor ferret was reduced to pH 5.7 at 3 dpi and was approximately 5.7 in all three contact ferrets upon first isolation (*Figure 8A*). Just as with Pair 1 viruses, airborne-transmitted P19 virus had an HA activation pH of 5.6 (*Figure 8A*), showing HA stabilization had occurred.

Next-generation sequencing showed the HA gene of P19 contained a higher frequency of variants than P24 (*Figure 8B*). Most variations remained minor and did not increase over time for Pair 2 viruses. For example, minor variants P24 NP-I363 and PB2-V661 from the original inoculum (*Figure 2D*) were enriched after one day of infection in donors but were not detected on day 3 and day 5 (*Figure 8—figure supplement 1A*). 2/3 of P24-infected contact ferrets acquired HA1-N16K mutations in the signal sequence on day 6, but this mutation was retained in only one ferret by day 8 with a decreased frequency (*Figure 8—figure supplement 1B*). All three P19-inoculated donors acquired a PB2-W98R mutation by day 3, but this mutation was less abundant on day 5 and did not

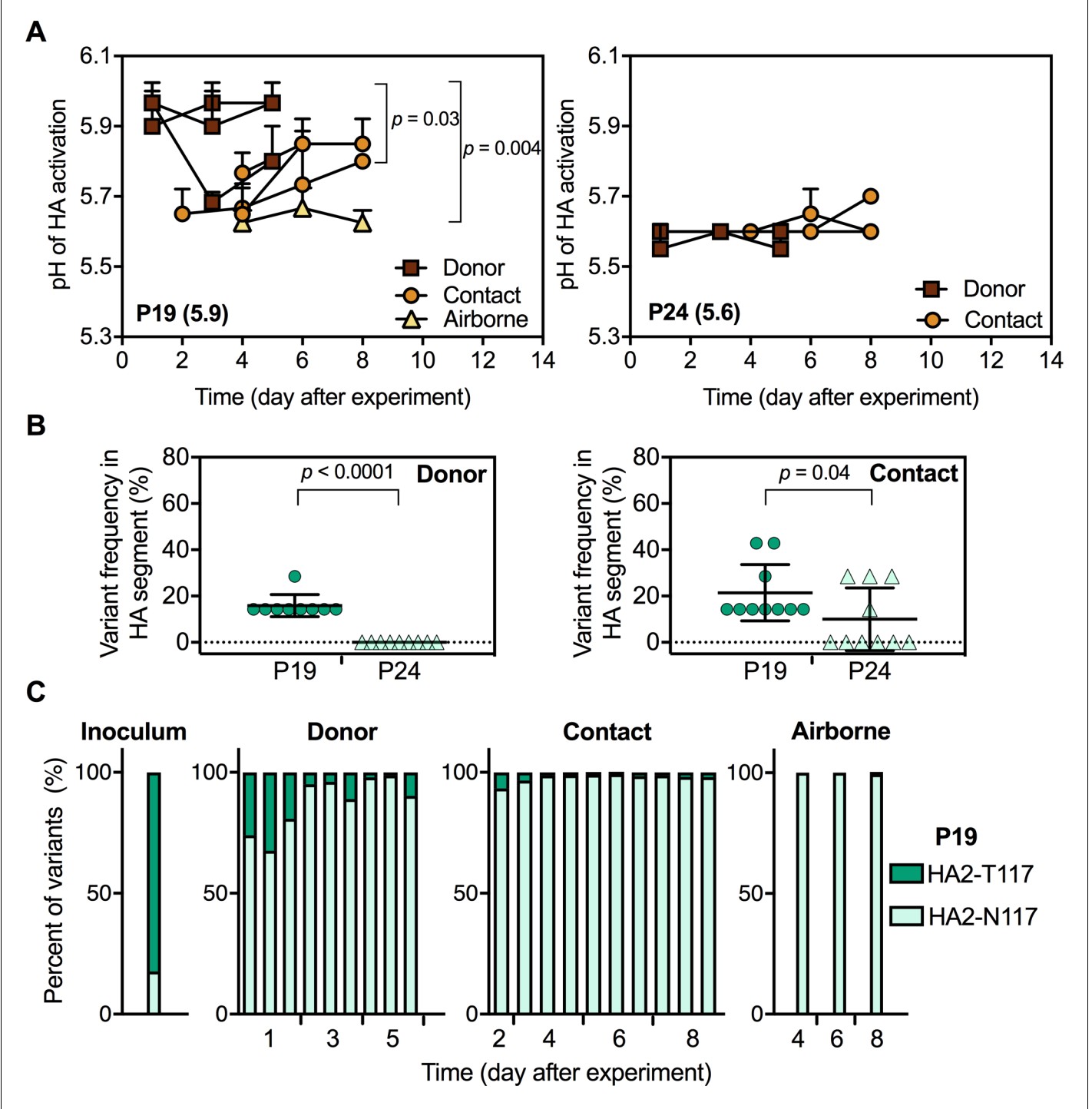

**Figure 8.** Phenotypes and genotypes of Pair 2 (P19 and P24) viruses after infection and transmission in ferrets. Virus phenotypes and genotypes were determined as described in *Figure 6*. (A) Virus HA activation pH values (means ± SD) measured by at least two independent syncytia assay experiments. *P* values were determined by one-way ANOVA followed by a Tukey's multiple comparisons test. (B) HA variant percentages in P19 and P24 viruses after infection and contact transmission in ferrets. *P* values were determined by Mann-Whitney U test. (C) Proportions of HA2-T117 and HA2-N117 in P19 before and after infection and transmission in ferrets.

The online version of this article includes the following figure supplement(s) for figure 8:

**Figure supplement 1.** Heat maps of mutants detected in nasal washes of Pair 2 infected and/or exposed ferrets during and after transmission.

carry forward to the three contact and one airborne ferret (*Figure 8—figure supplement 1C–E*). As described above, the inoculum of P19 contained 83% HA2-T117 and 17% HA2-N117. HA2-N117 outgrew HA2-T117 in all three inoculated ferrets within 1 day (*Figure 8C*, *Figure 8—figure supplement 1C*). HA activation pH values for 2/3 of the P19 donor ferrets remained at pH 5.9 despite containing a majority of the stabilizing HA2-N117 variant. High-pH syncytia formation for these samples was most likely induced by the remaining minority population of the HA2-T117 variant. Upon first detection in P19 contact ferrets, HA2-N117 was over 90% abundant, and this variant comprised over 99% of the population in the airborne ferret. Contamination by P24 (also contained HA2-N117) was ruled out, as HA2-N117-containing P19-group samples retained identifying nucleotide PA-1749T (P24 had nucleotide PA-1749C at this position). Similar to Pair 1, the dominant population of P19 in the inoculum with an unstable HA protein (pH 5.9, HA2-T117) was outcompeted and did not airborne transmit. Although airborne transmission in the Pair 2 experiment was infrequent (1/6 total), a moderately stable HA (pH 5.6, HA2-N117) was necessary for efficient replication and airborne transmission.

## Discussion

This work showed that HA proteins from pandemic-clade human and swine IAVs are relatively stable (activation pH 5.0–5.5) while those from swine gamma viruses are less stable (pH 5.5–5.9). A moderately stable HA protein (pH 5.5–5.6) was necessary for efficient replication and partial airborne transmission of swine gamma viruses in ferrets. Despite replicating to higher levels in MDCK cells than human pandemic viruses, swine gamma viruses with unstable HA proteins (pH 5.8–5.9) were not airborne transmitted and were outgrown quickly by minor variants that contained stabilized HA proteins. Overall, these results are consistent with the hypothesis that HA stabilization is necessary for airborne transmission of swine IAVs in ferrets. The results also suggest that a stabilized HA (pH <5.6) and human-like receptor-binding in tandem are necessary for IAV airborne transmissibility in ferrets.

The HA protein has two primary roles during virus entry: receptor binding and acid-induced membrane fusion. Previous studies have indicated HA binding to $\alpha-2,6$-linked sialic acid receptors and a relatively low HA activation pH may contribute to airborne transmissibility in ferrets (reviewed in *de Graaf and Fouchier, 2014*; *Cauldwell et al., 2014*; *Russell et al., 2018*), a phenotype often considered a predictor of human adaptation. In the present study, ferrets were infected with natural isolates of swine H1N1 gamma viruses that varied in HA stability and had dual affinities toward $\alpha-2,6-$ and $\alpha-2,3$-linked sialic acid receptors. Viruses containing an unstable HA protein did not transmit by the airborne route, and all viruses isolated from airborne recipients contained stabilized HA proteins. For A/H1N1pdm, a destabilizing mutation (HA1-Y17H, pH 6.0) eliminated airborne transmissibility in ferrets despite the virus retaining $\alpha-2,6$ receptor-binding specificity (*Russier et al., 2016*). This suggested that $\alpha-2,6$ receptor-binding specificity was insufficient to promote airborne transmission in the absence of a stabilized HA protein. The mutant virus regained airborne transmissibility by acquiring two mutations that lowered its activation pH to 5.3. A/H5N1 viruses with relatively unstable HA proteins (pH >5.5) were engineered to have $\alpha-2,6$ receptor-binding specificity but did not become airborne transmissible in ferrets until acquiring a mutation that decreased the HA activation pH to 5.5 or lower (*Imai et al., 2012*; *Herfst et al., 2012*; *Linster et al., 2014*). In other studies, HA stabilization of A/H5N1 viruses in the absence of $\alpha-2,6$ receptor-binding specificity increased virus growth in the upper respiratory tract of ferrets but was insufficient to promote airborne transmissibility (*Zaraket et al., 2013*; *Shelton et al., 2013*). Altogether, these results suggest $\alpha-2,6$ receptor-binding and a stabilized HA protein (pH <5.6) in tandem are necessary for IAV airborne transmission in ferrets; neither property appears sufficient without the other.

For the swine gamma viruses studied here, $\alpha-2,6$ receptor binding and a relatively stable HA protein was necessary for airborne transmissibility in ferrets but was insufficient for robust transmission. The swine gamma viruses also bound $\alpha-2,3$ receptors, which was associated with a loss of airborne transmissibility for A/H1N1/1918 in ferrets (*Tumpey et al., 2007*), perhaps due to increased binding to human bronchial mucins (*Lamblin et al., 2001*; *Vahey and Fletcher, 2019*). A/H7N9 viruses with dual receptor-binding affinities have also been shown to have limited or very poor airborne transmissibility (*Richard et al., 2013*; *Belser et al., 2013*), although these viruses may lack other properties necessary for airborne transmissibility such as a stabilized HA protein (*Zaraket et al., 2015*). Ferrets, humans, and other animals differentially express complex

glycopolymers (*Walther et al., 2013*; *Jia et al., 2014*; *Byrd-Leotis et al., 2014*), and mammalian and avian influenza viruses bind differently to both sialylated and phosphorylated, nonsialylated glycans (*Byrd-Leotis et al., 2019*). Thus, refinements in binding to $\alpha-2,6$-linked sialic acid (and perhaps other) receptors may be needed to enhance airborne transmissibility. For example, airborne transmission by the A/H1N1pdm09 virus was conferred by binding to long-chain $\alpha-2,6$-linked sialic acids even when $\alpha-2,3$ binding was retained (*Lakdawala et al., 2015*).

In addition to HA receptor binding and stability, other adaptive properties contributing to influenza virus transmission include polymerase activity, virus morphology, and resistance to host-based restrictions to replication (*Mänz et al., 2012*; *Bogdanow et al., 2019*; *Long et al., 2019a*). Such adaptations may also be needed for swine gamma viruses to adapt highly efficient airborne transmissibility in ferrets. In this study, only 1/3 of airborne-exposed ferrets in the P4 group seroconverted while G15-HA1-S210 airborne-transmitted 3/3. Both P4 and G15-HA1-S210 had identical HA sequences (including HA1-S210) and HA stabilities. As the two viruses differed only at amino-acid residues PA-271 and PB2-648, suboptimal polymerase activity is most likely responsible for the limited airborne transmissibility of P4.

Additional supporting data from follow-up studies could support the conclusion of this study that HA stabilization is necessary but not sufficient for airborne transmission of swine gamma viruses between ferrets. Samples obtained from nasal washes in this experiment could be plaque purified to isolate G15-HA1-N210, G15-HA1-S210, and P4-HA1-S210. G15-HA1-N210 and G15-HA1-S210 are identical except for the stability-altering residue at HA1-210; therefore, differences in replication and transmission in ferrets between the two viruses would map exclusively to the HA protein. Comparing infection with G15-HA1-S210 and P4-HA1-S210, which have identical HA genes but differ at polymerase residues PA-271 and PB2-648, may reveal the importance of the polymerase genes in supporting airborne transmission of the related swine gamma viruses.

Swine are susceptible to both avian and human IAVs, in part by supporting replication of viruses that bind to either $\alpha-2,3$- or $\alpha-2,6$-linked receptors. We speculate swine may also serve as an avian-human intermediate host for HA stabilization. H1N1 viruses of the classical, Eurasian, and triple-reassortant swine lineages isolated before 2009 have HA activation pH values of 5.4–6.0 (*Russier et al., 2016*). The present study showed that gamma-clade IAVs isolated 2012–2016 also have moderately stable to unstable HA proteins (pH 5.5–6.0) and that a moderately stable HA protein (pH 5.5–5.6) is necessary for airborne transmission in ferrets. Recently, three H1N1 variant (H1N1v) viruses of the gamma clade that had been isolated from humans were found to have HA activation pH values of ~5.7 (*Pulit-Penaloza et al., 2018*). Moreover, an alpha-lineage H1N2v virus with an HA activation pH of 5.5 promoted airborne transmission in ferrets while delta-clade H1N2 viruses with less stable HA proteins (pH 5.6–5.7) promoted only partial airborne transmission (1 or 2 out of 3) (*Pulit-Penaloza et al., 2018*). This supports the notion that increased HA stability promotes increased airborne transmissibility in ferrets. As the transmitted viruses collected from airborne recipients in the above study were not phenotyped and genotyped (*Pulit-Penaloza et al., 2018*), it is possible that the transmitted H1N1v and H1N2v viruses had activation pH values differing from input virus, similar to our results of gamma viruses G15 and P19.

In summary, contemporary swine H1N1 gamma viruses were found to have HA activation pH values higher than those of the currently circulating pandemic-clade, and HA stabilization enhanced gamma virus growth and was necessary but not sufficient for airborne transmission in ferrets. This work, combined with other recent studies (*Imai et al., 2012*; *Herfst et al., 2012*; *Linster et al., 2014*; *Russier et al., 2016*; *Pulit-Penaloza et al., 2018*), support the proposition that HA stability in tandem with receptor-binding specificity should be considered in pandemic risk assessment tools. Gamma viruses are one of the most prevalent lineages in swine (*Diaz et al., 2017*) and are widely distributed in many locations including USA, Mexico, China, Hong Kong, and South Korea (*Anderson et al., 2016*; *Nelson et al., 2015*). These viruses have caused sporadic human infections in recent years (*World Health Organization (WHO), 2018*). Prior studies suggest IAVs endemic in swine reassort readily with 2009 pandemic viruses and many have obtained pandemic internal genes (*Rajão et al., 2017*; *Ducatez et al., 2011*; *Vijaykrishna et al., 2010*). In this study, we found viruses G15 and P4 contained M and NP genes of 2009 human pandemic origin, while P19 and P24 contained M, PB2, PB1, and PA of 2009 human pandemic origin. The harboring of pandemic gene segments by swine gamma viruses increases their potential to evolve into a pandemic virus. For A/H3N2 swine viruses, reassortment with 2009 pandemic genes has been found to be associated with

increased pathogenicity (*Rajão et al., 2017*). We conclude that the evolution of currently circulating swine IAVs, especially gamma clade, should be closely monitored for increased pandemic potential.

# Materials and methods

## Key resources table

| Reagent type (species) or resource | Designation | Source or reference | Identifiers | Additional information |
|---|---|---|---|---|
| Animal (*Mustela putorius furo*) | Fitch ferret | Triple F Farms | | male, 5–6 months old |
| Cell line (*Canis lupus familiaris*) | MDCK, Madin-Darby Canine Kidney epithelial | ATCC | ATCC Cat# CCL-34, RRID:CVCL_0422 | |
| Cell line (*Chlorocebus sabaeus*) | Vero, African green monkey kidney epithelial | ATCC | ATCC Cat# CCL-81, RRID:CVCL_0059 | |
| Cell line (*Sus scrofa*) | Swine testis (ST) fibroblast | ATCC | ATCC Cat# CRL-1746, RRID:CVCL_2204 | |
| Cell line (*Homo sapiens*) | 293T/17 [HEK 293T/17], Human epithelial kidney expressing SV40 large T antigen | ATCC | ATCC Cat# CRL-11268, RRID:CVCL_1926 | |
| Recombinant DNA reagent | pHW2000 A/TN/2009 cDNA reverse genetics plasmids | *Russier et al., 2016* | PMID:26811446 | |
| Recombinant DNA reagent | pHW2000 swine gamma HA and NA cDNA reverse genetics plasmids | This paper | | |
| Software | CLC Genomics Workbench 11.0.1 | Qiagen | | |
| Software | Prism 7 | GraphPad | | |
| Strain, strain background (Influenza A virus) | A/Tennessee/1-560/2009 (H1N1) | NCBI | NCBI:txid646491 | |
| Strain, strain background (Influenza A virus) | A/swine/Illinois/2A-1213-G15/2013 (H1N1) HA | Genbank | MT533249 | A_sw_IL_1213G15_13_HA |
| Strain, strain background (Influenza A virus) | A/swine/Illinois/2B-0314-P4/2014 (H1N1) HA | Genbank | MT533251 | A_sw_IL_2B0314P4_14_HA |
| Strain, strain background (Influenza A virus) | A/swine/Illinois/2E-0113-P19/2013 (H1N1) HA | Genbank | MT533255 | A_sw_IL_2E0113P19_13_HA |
| Strain, strain background (Influenza A virus) | A/swine/Illinois/2E-0113-P24/2013 (H1N1) HA | Genbank | MT533252 | A_sw_IL_2E0113P24_13_HA |

## Cells and viruses

Madin-Darby canine kidney (MDCK) cells, African green monkey kidney (Vero) cells, swine testicular (ST) cells, and human embryonic kidney (HEK) 293 T cells were maintained in minimum essential medium (MEM, Thermo Fisher Scientific), Dulbecco's Modified Eagle Medium (DMEM, Life Technologies), DMEM, and Opti-MEM reduced Serum Medium (Life Technologies), respectively. All culturing media were supplemented with 10% HyClone standard fetal bovine serum (FBS, Life Technologies) and 1% penicillin/streptomycin (P/S, Thermo Fisher Scientific). All cells were grown at

37°C with 5% $CO_2$ (*Hu et al., 2016*; *Hu et al., 2017a*). Cell lines were authenticated by short-tandem repeat profiling and tested free of mycoplasma contamination.

Fifty-five H1N1 influenza A viruses (IAVs) (2009–2016) were propagated in MDCK cells less than three passages with 1 μg/ml tosylsulfonyl phenylalanyl chloromethyl ketone (TPCK)-treated trypsin, as described previously (*Hu et al., 2016*). The four viruses used in ferret experiments (G15, P4, P19, and P24) were recovered from swine and then passaged two times in MDCK cells before the experiments reported here. Recombinant viruses used in this study were generated using reverse genetics using the HA and NA genes from the indicated viruses along with internal genes from A/TN/2009, and were propagated in MDCK cells less than three passages (*Hu et al., 2016*; *Hu et al., 2017b*; *Russier et al., 2016*).

## Phylogenic analysis of virus HA segments

Full-length HA sequences of viruses used in this study (2009–2016) were obtained by Sanger sequencing, retrieved from St. Jude Children's Research Hospital repository, or GenBank (https://www.ncbi.nlm.nih.gov/genbank/). Additional full-length HA protein sequences of human and swine influenza A viruses (1918 ~ 2016) were retrieved from GenBank. Multiple sequence alignments (MSA) were generated by using MAFFT v7.273 (*Katoh and Standley, 2013*). The phylogenetic analysis was performed using a maximum-likelihood tree to represent the evolutionary relationship among different strains, as described elsewhere (*Li et al., 2018*). The phylogenetic trees were inferred using a maximum-likelihood method implemented in RAxML v8.2.9 (*Stamatakis, 2014*). A GAMMA model of rate heterogeneity and a generalized time-reversible (GTR) substitution model were applied in the analyses. Phylogenetic trees were then visualized by ggtree v1.6.11 (*Yu et al., 2017*).

## Virus growth assay

Tested viruses were inoculated into MDCK cells or ST cells in six-well plates at a multiplicity (MOI) of 0.01 PFU/cell. The inocula were removed after 1 hr incubation at 37°C. Infected cells were washed by PBS (pH ~7.4) twice and covered by 3 ml MEM culture media supplemented with 1 μg/ml TPCK-treated trypsin. Infected cells were then maintained at 37°C. At indicated time points, 300 μl of cell culture supernatant was collected and titrated by $TCID_{50}$ on MDCK cells (*Hu et al., 2017a*; *Russier et al., 2016*).

## HA stability

HA stability was measured using syncytia and/or virus inactivation assays. In the syncytia assay, Vero cells in 24-well plates were infected at an MOI of 3 PFU/cell. At 16 hr post-infection (hpi), infected cells were treated with DMEM supplemented with 5 μg/ml TPCK-treated trypsin for 15 min. Subsequently, infected cells were maintained with pH-adjusted PBS buffers ranging from 4.8 to 6.2 for 15 min. After aspiration of pH-adjusted PBS, infected cells were incubated in DMEM supplemented with 5% FBS for 3 hr at 37°C. The cells were then fixed and stained using a Hema 3 Fixative and Solutions (Fisher Scientific). Photomicrographs of cells containing or lacking syncytia were recorded using a light microscope (*Russier et al., 2016*; *Reed et al., 2010*). A baseline of no virus-induced syncytia formation was obtained by exposing the cells to pH 6.2 media, a condition under which Vero cells formed no visible syncytia. Micrographs were scored positive for syncytia formation if a field contained at least two syncytia that had at least five nuclei. HA activation pH values for syncytia assays were reported as the highest pH that induced syncytia as judged by positive scoring. Differences in HA cell-surface expression could affect the number of syncytia observed per field but would not be expected to alter the reported threshold pH level at which syncytia are induced.

To perform the virus inactivation assay, tested viruses were mixed with pH-adjusted PBS buffers in a ratio of 1:100 and incubated at 37°C for 1 hr. After that, the resulting infectious viruses were titrated by $TCID_{50}$ on MDCK cells (*Russier et al., 2016*; *Zaraket et al., 2013*).

## Hemagglutinin (HA) assay

Serial two-fold dilutions of tested viruses were prepared in a U-shaped 96-well plate (50 μl/well). PBS was used as a negative control (50 μl/well). After that, 50 μl of 0.5% of turkey red blood cells (tRBCs) was added into each well and mixed gently. The plate was then incubated at room

temperature for 30 min. The HA titers of the tested viruses were defined as the highest dilutions that tRBC clumping occurred in the bottom of the wells (*Webster et al., 2002*).

## Solid-phase receptor binding assay

Virus receptor-binding specificity was evaluated using a solid-phase receptor binding assay (*Zaraket et al., 2015*; *Russier et al., 2016*). 50 µl of tested viruses (32 HA Units) were inoculated into 96-well plates coated with fetuin. After overnight incubation at 4°C, inocula were removed. The plates were washed by PBS five times and blocked with 200 µl of PBS buffer with 0.1% desialylated bovine serum albumin (Sigma) for 2 hr. The plates were then washed by PBS three times and covered with 50 µl of indicated sialylglycopolymers. After incubation 1 hr at 4°C, the sialylglycopolymers were replaced with 50 µl of horseradish peroxidase (HRP)-conjugated streptavidin. The plates were maintained at 4°C for another 1 hr. After that, 50 µl of TMB substrate solution (Thermo Scientific) were added into each well, and the plates were incubated at room temperature for 30 min. Finally, 50 µl of 3% $H_2SO_4$ were added and the plates were read at 450 nm to obtain absorbance values on BioTek microplate reader (BioTek Instruments, Inc).

## Virus genome sequencing and analyses

Virus whole-genome sequences were obtained by next-generation sequencing (*Russier et al., 2017*; *Zaraket et al., 2015*). Viral RNAs were extracted from ferret nasal wash samples (unless otherwise indicated). Subsequently, virus cDNAs were obtained by reverse-transcription PCR using SuperScript III First-strand Synthesis System (Thermo Fisher Scientific). DNA libraries were prepared by PCR amplification using Phusion High-Fidelity PCR Master Mix with HF Buffer (New England BioLabs). The resulting PCR products were purified by QIAquick Gel Extraction Kit (Qiagen) and submitted to the St. Jude Harwell Center for virus whole-genome sequencing. Sequencing data were analyzed using CLC Genomics Workbench version 11.0.1. Variants were reported when the predefined quality scores were met, and they were presented in forward and reverse reads at an equal ratio. All variants presented in this study were supported by at least 10 reads with a minimum frequency of 5%. Heat maps and parts of whole were generated by using GraphPad Prism version 7 software (GraphPad Software, San Diego, CA).

## Ferret transmission experiments

Thirty-six male Fitch ferrets (five- to six-month-old, purchased from Triple F Farms, Sayre, PA) were verified to be serologically negative for currently circulating IAVs, influenza B viruses, and viruses used in this study by HAI assay. Viruses A/swine/Illinois/2A-1213-G15/2013 (G15), A/swine/Illinois/2B-0314-P4/2014 (P4), A/swine/Illinois/2E-0113-P19/2013 (P19) and A/swine/Illinois/2E-0113-P24/2013 (P24) were used for ferret experiments. On day 0, 3 naive ferrets each (Donor) were intranasally inoculated with $10^6$ PFU of tested viruses in 500 µl and were caged separately. On day 1, another 3 naive ferrets each (Contact) were co-housed with the Donor ferrets. At the same time, 3 naive ferrets each (Airborne) were introduced in adjacent cages. Aerosol exposure was permitted among ferrets within tested viruses but was not allowed between the tested viruses. Ferret body weight and temperature were monitored daily until day 14. Ferret nasal washes were collected every other day until day 14. Additionally, all ferrets were euthanized on day 21 to collect whole blood. Extracted ferret sera were subjected to HAI assay (*Zaraket et al., 2015*; *Russier et al., 2016*). During the experiments, temperature and humidity in each cubicle were set and controlled to 22°C and 40–60%, respectively. Animal studies were performed in compliance with St. Jude Children's Research Hospital Animal Care and Use Committee guidelines.

## Hemagglutinin inhibition (HAI) assay

Ferret sera were collected and treated with receptor-destroying enzymes (Denka Seiken Co., LTD). After incubation at 37°C overnight and treatment at 56°C for 30 min, the resulting samples were examined with 0.5% TRBCs. The HAI titers were defined as the highest dilutions that completely inhibited hemagglutinin (*Zaraket et al., 2015*).

## Statistical analysis

All data analyses were performed using GraphPad Prism version 7. One-way ANOVA followed by a Tukey's multiple comparisons test, Mann-Whitney U test, or two-tailed Student's $t$-test were used to determine statistical significances. $P$ values $< 0.05$ were considered significant.

## Acknowledgements

We thank Shannon Moore for help with cloning and Kimberly Friedman and David Walker for providing viruses and sequence data. We thank the St. Jude Animal Resources Center (ARC) for help with the ferret experiments. We also appreciate assistance by the Hartwell Center DNA Sequencing and Genotyping, Hartwell Center Functional Genomics, and the Hartwell Center Genome Sequencing Facility. This work was funded, in part, by the National Institute of Allergy and Infectious Diseases under Centers of Excellence for Influenza Research and Surveillance (CEIRS) contract no. HHSN272201400006C, St. Jude Children's Research Hospital, and the American Lebanese Syrian Associated Charities (ALSAC). The content is solely the responsibility of the authors and does not necessarily represent the official views of the National Institutes of Health.

## Additional information

### Funding

| Funder | Grant reference number | Author |
|---|---|---|
| National Institute of Allergy and Infectious Diseases | HHSN272201400006C | Meng Hu<br>Guohua Yang<br>Jennifer DeBeauchamp<br>Jeri Carol Crumpton<br>Hyunsuh Kim<br>Lisa Kercher<br>Andrew S Bowman<br>Robert G Webster<br>Richard J Webby<br>Charles J Russell |
| St. Jude Children's Research Hospital | | Meng Hu<br>Guohua Yang<br>Jennifer DeBeauchamp<br>Jeri Carol Crumpton<br>Hyunsuh Kim<br>Lisa Kercher<br>Robert G Webster<br>Richard J Webby<br>Charles J Russell |
| American Lebanese Syrian Associated Charities | | Meng Hu<br>Guohua Yang<br>Jennifer DeBeauchamp<br>Jeri Carol Crumpton<br>Hyunsuh Kim<br>Lisa Kercher<br>Robert G Webster<br>Richard J Webby<br>Charles J Russell |
| National Institute of Allergy and Infectious Diseases | R01AI116744 | Lei Li<br>Xiu-Feng Wan |

The funders had no role in study design, data collection and interpretation, or the decision to submit the work for publication.

### Author contributions

Meng Hu, Conceptualization, Data curation, Formal analysis, Supervision, Methodology, Writing - original draft, Project administration, Writing - review and editing; Guohua Yang, Jennifer DeBeauchamp, Jeri Carol Crumpton, Investigation; Hyunsuh Kim, Methodology; Lei Li, Investigation, Methodology; Xiu-Feng Wan, Investigation, Methodology, Writing - review and editing; Lisa Kercher,

Supervision, Methodology; Andrew S Bowman, Resources; Robert G Webster, Resources, Writing - review and editing; Richard J Webby, Resources, Funding acquisition, Writing - review and editing; Charles J Russell, Conceptualization, Data curation, Formal analysis, Supervision, Funding acquisition, Methodology, Writing - original draft, Project administration, Writing - review and editing

**Author ORCIDs**
Charles J Russell  https://orcid.org/0000-0001-5683-3990

**Ethics**
Animal experimentation: Animal studies were performed in compliance with St. Jude Children's Research Hospital Animal Care and Use Committee guidelines under reviewed ACUC protocol 459.

**Decision letter and Author response**
Decision letter https://doi.org/10.7554/eLife.56236.sa1
Author response https://doi.org/10.7554/eLife.56236.sa2

---

## Additional files

**Supplementary files**
• Supplementary file 1. Supplementary Data.
• Transparent reporting form

**Data availability**
All data generated or analysed during this study are included in the manuscript and supporting files.

---

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
