## [Decision Letter]

**Acceptance summary:**

The haemagglutinin (HA) envelope glycoprotein of influenza A viruses mediates receptor-binding and membrane fusion. The characteristics of these proteins that promote airborne transmission of influenza viruses are not fully established. This paper presents data supporting the notion that haemagglutinins requiring more acidic pH for fusion, i.e. that are more stable, promote airborne transmission. The authors argue that this feature should be incorporated into pre-pandemic assessments.

**Decision letter after peer review:**

Thank you for submitting your article "HA stabilization promotes replication and transmission of swine H1N1 gamma influenza viruses in ferrets" for consideration by *eLife*. Your article has been reviewed by three peer reviewers, and the evaluation has been overseen by a Reviewing Editor and Karla Kirkegaard as the Senior Editor. The reviewers have opted to remain anonymous.

The reviewers have discussed the reviews with one another and the Reviewing Editor has drafted this decision to help you prepare a revised submission.

Essential revisions:

Based on the individual reviews below and discussions between the reviewers, we will be willing to consider a revised version of the paper that addresses the issues raised in the reviews. The most significant of these being a lack of statistical analysis, and requests for more detail on some of the experiments as well as the fusion assay. One reviewer is particularly concerned that your hypothesis is only supported by a very detailed examination of the properties of the recovered material in one of the two experimental transmissions and that the hypothesis has not been tested further. He/she is concerned that the case for HA stabilization promoting replication and transmission is overstated. You should seek to address this issue in a revised paper.

Reviewer #1:

Hu et al. have analysed the properties of swine γ clade and pandemic flu viruses to determine whether specific properties of the hemagglutinin molecule (HA), namely receptor specificity and fusion pH, influence transmission in ferrets. Based on the data presented, the authors conclude that both properties are important and that viruses with more stable HA molecules (i.e. requiring lower pH for fusion) are better able to undergo airborne transmission in ferrets. From this analysis, and similar conclusions from other published work, the authors argue that HA stability should be included as a risk factor when considering the pandemic risk of new flu viruses.

Overall, the work is well done, though it might be argued that the animal data is somewhat compromised by the small sample size. Nevertheless, the conclusions are consistent with those published from other laboratories, using different flu viruses. I have just a few remarks for the authors to consider.

1) The conclusion that viruses with more stable HAs undergo airborne transmission more efficiently is undermined with the P4/G15 pair, in which 3/3 G15 challenges showed transmission, but only 1/3 P4 airborne challenges sero-converted. Although the authors show the virus transmitted in the G15 sample was the minority 210S virus, they do not really discuss why the P4 result was lower than might have been expected.

2) The authors should give more details of the syncytium assay used to determine the fusion pH for the different viruses. Do they score a fusion index for the different pH conditions and, if so, what is the threshold at which fusion is deemed to have occurred? Is HA expressed at similar levels for all of the viruses and, if not, would this comprise fusion efficiency?

Reviewer #2:

This manuscript addresses a very important question that remains a critical area of influenza biology that is needed for assessing the risks associated with emerging influenza viruses: The virological parameters promote airborne transmission of influenza viruses is not well understood. In this manuscript this has been explored for a subset of swine influenza viruses, those in a subgroup of the swine H1N1 viruses in the γ clade. These γ clade wine influenza viruses have caused zoonotic infections and they provided six of the eight RNA segments of the H1N1pdm09 viruses. In this sense the material is suitable for publication in *eLife*.

The hypothesis is that viruses able to be spread by an airborne route have hemagglutinin glycoproteins (HA) that undergo the low pH-induced conformational change of the HA required for fusion of the virus envelope with the membranes of the cell at a reduced pH. At first examination this was not apparent from the simple phenotypes of the viruses chosen for study but a detailed examination of the viruses supports this hypothesis.

1) The results compared the low pH induced show that human H1N1pdm09 viruses and swine viruses with the HA derived from the human virus displayed a lower pH of fusion of infected vero cells than did the swine viruses chosen to exemplify the swine gamma lineage viruses. The authors then chose examples of swine viruses to examine in more detail, two pairs being chosen that showed differences in their pH of fusion within the pair.

Receptor binding, fusion and replication properties of these viruses were established and then examined in transmission studies between infected ferrets, direct contact ferrets and ferrets infected by an airborne route.

2) At first examination of the results seemed not to be consistent with the hypothesis. What was seen was that in Pair 1 seemingly more efficient transmission in the virus showing a reduced pH of fusion was not apparent from the simple phenotypes of the viruses chosen for study. The results (Figure 5) show from pair 1 virus G15 showed better or earlier infection of the ferrets by an airborne route than did virus P4, notably as assessed by serology. Whether these results are statistically significant differences need to be established. In Figure 7 the results of the ferret transmission from the other pair of viruses is shown. Here virus P24, with the lower pH of fusion, showed now evidence for transmission in ferrets by the airborne route whilst P19 resulted in transmission to one of the three ferrets exposed by the airborne route. No statistical analysis were presented. In both cases the virus with the higher pH of fusion was the one that seemed to transmit by the airborne route more efficiently – taken at face value.

3) However, more work was done to examine the properties of the virus that was able to be transmitted by the airborne route. This was done for virus recovered from the P19/P24 pair. Here the virus collected from donor ferrets, contact ferrets and airborne infected ferrets were examined. The results shown in Figure 8 seemed to show that virus form the contact-infected and the airborne-infected ferrets from the donor infected with P19 showed a lower pH of fusion than seen in some of the donor ferrets whilst this drop in pH of fusion was not seen in the directly contact ferrets infected from the P24 infected ferrets. The results shown in Figure 8C are consistent with the proposal that transmission might be linked to a polymorphism at residue 117 of HA2. However, selection of N117 of HA2 was observed in the donor ferrets. The selection of the variant was therefore not clear cut.

4) Overall, the hypothesis that a low pH of fusion promotes virus transmission is supported by the work described when examined in detail. The hypothesis is not supported without the very detailed examination of the properties of the recovered material from the infected ferrets in one of the two experimental transmissions. However, no further testing of the authors' hypothesis has been done on the recovered viruses to examine their transmission between ferrets.

Reviewer #3:

The manuscript by Hu et al. describes studies on gamma clade H1N1 swine influenza A viruses and the genetic and phenotypic changes that allow for airborne transmission in ferrets, which are the accepted "gold standard" animal model for human transmission. The naturally circulating gamma clade swine viruses have hemagglutinin (HA) glycoproteins that generally mediate membrane fusion at relatively high pH compared to human strains and other swine strains. Comparison of ferret transmission phenotypes of virus pairs with similar genetic backbones showed that single genetic differences that lead to more stable HAs (lower fusion pH) correlated with increased airborne transmission. Notably, the increased transmission observed in one example could be traced directly to selection of a stabilizing HA change detected in the virus population of airborne-transmitted recipient ferrets. Other phenotypic changes including receptor binding specificity were not observed when comparing virus pairs. The results provide direct evidence to support the premise that HA stability serves as a potential determinant of airborne transmission and should be included in risk assessment tools and guidelines for determining pandemic potential of influenza viruses circulating in swine or other non-human hosts. The experiments are well designed and the data are solid and support the conclusions made by the investigators. The paper is well written and will be of interest to a broad readership.

---

## [Author Response]

Essential revisions:Based on the individual reviews below and discussions between the reviewers, we will be willing to consider a revised version of the paper that addresses the issues raised in the reviews. The most significant of these being a lack of statistical analysis,

We added the suggested statistical analyses to the Results section and the figure legends as described below to reviewer 2’s comments.

Requests for more detail on some of the experiments as well as the fusion assay.

We addressed these comments below.

One reviewer is particularly concerned that your hypothesis is only supported by a very detailed examination of the properties of the recovered material in one of the two experimental transmissions and that the hypothesis has not been tested further. He/she is concerned that the case for HA stabilization promoting replication and transmission is overstated. You should seek to address this issue in a revised paper.

We appreciate the reviewers’ opinions and the COVID-19 policy implemented by *eLife*. In the revised manuscript, we both limit the claims and explicitly state that the relevant conclusion requires additional supporting data. In the future when we are able, we intend to perform the recommended follow-up experiment and write a brief paper that may support the conclusions of this paper.

Limiting the claims:

1) We edited the end of the Abstract to, “These results suggest (instead of show) swine influenza viruses containing both a stabilized HA and α-2,6 receptor binding in tandem pose greater pandemic risk. Increasing evidence supports adding HA stability to pre-pandemic risk assessment algorithms.”

2) We deleted the last sentence of the Introduction that stated: “Indeed, the results confirmed HA stabilization was necessary for γ virus replication and transmission in ferrets.”

3) We edited the first paragraph of the Discussion to, “Overall, these results are consistent with (instead of confirmed) the hypothesis that HA stabilization is necessary for airborne transmission of swine IAVs in ferrets.”

4) We added the following caveat to the first sentence of the last paragraph in the Discussion, “In summary, contemporary gamma swine viruses were found to have HA activation pH values higher than those of the currently circulating pandemic-clade, and HA stabilization enhanced gamma virus growth and was necessary but not sufficient for airborne transmission in ferrets.”

Additional data that could support the conclusion:

We added the following paragraph to the Discussion: “Additional supporting data from follow-up studies could support the conclusion of this study that HA stabilization is necessary but not sufficient for airborne transmission of swine gamma viruses between ferrets. Samples obtained from nasal washes in this experiment could be plaque purified to isolate G15-HA1-N210, G15-HA1-S210, and P4-HA1-S210. G15-HA1-N210 and G15-HA1-S210 are identical except for the stability-altering residue at HA1-210; therefore, differences in replication and transmission in ferrets between the two viruses would map exclusively to the HA protein. Comparing infection with G15-HA1-S210 and P4-HA1-S210, which have identical HA genes but differ at polymerase residues PA-271 and PB2-648, may reveal the importance of the polymerase genes in supporting airborne transmission of the related swine gamma viruses.”

Reviewer #1:Hu et al. have analysed the properties of swine gamma clade and pandemic flu viruses to determine whether specific properties of the hemagglutinin molecule (HA), namely receptor specificity and fusion pH, influence transmission in ferrets. Based on the data presented, the authors conclude that both properties are important and that viruses with more stable HA molecules (i.e. requiring lower pH for fusion) are better able to undergo airborne transmission in ferrets. From this analysis, and similar conclusions from other published work, the authors argue that HA stability should be included as a risk factor when considering the pandemic risk of new flu viruses.Overall, the work is well done, though it might be argued that the animal data is somewhat compromised by the small sample size. Nevertheless, the conclusions are consistent with those published from other laboratories, using different flu viruses. I have just a few remarks for the authors to consider.1) The conclusion that viruses with more stable HAs undergo airborne transmission more efficiently is undermined with the P4/G15 pair, in which 3/3 G15 challenges showed transmission, but only 1/3 P4 airborne challenges sero-converted. Although the authors show the virus transmitted in the G15 sample was the minority 210S virus, they do not really discuss why the P4 result was lower than might have been expected.

In the fourth paragraph of the Discussion, we added the following two sentences to discuss the P4 results: “In this study, only 1/3 of airborne-exposed ferrets in the P4 group seroconverted while G15-HA1-S210 airborne-transmitted 3/3. Both P4 and G15-HA1-S210 had identical HA sequences (including HA1-S210) and HA stabilities. As the two viruses differed only at amino-acid residues PA-271 and PB2-648, suboptimal polymerase activity is most likely responsible for the limited airborne transmissibility of P4.”

2) The authors should give more details of the syncytium assay used to determine the fusion pH for the different viruses. Do they score a fusion index for the different pH conditions and, if so, what is the threshold at which fusion is deemed to have occurred? Is HA expressed at similar levels for all of the viruses and, if not, would this comprise fusion efficiency?

We added the following explanation to the “HA stability” section of the Materials and methods: “A baseline of no virus-induced syncytia formation was obtained by exposing the cells to pH 6.2 media, a condition under which Vero cells formed no visible syncytia. Micrographs were scored positive for syncytia formation if a field contained at least two syncytia that had at least 5 nuclei. HA activation pH values for syncytia assays were reported as the highest pH that induced syncytia as judged by positive scoring. Differences in HA cell-surface expression could affect the number of syncytia observed per field but would not be expected to alter the reported threshold pH level at which syncytia are induced.”

Reviewer #2:This manuscript addresses a very important question that remains a critical area of influenza biology that is needed for assessing the risks associated with emerging influenza viruses: The virological parameters promote airborne transmission of influenza viruses is not well understood. In this manuscript this has been explored for a subset of swine influenza viruses, those in a subgroup of the swine H1N1 viruses in the γ clade. These γ clade wine influenza viruses have caused zoonotic infections and they provided six of the eight RNA segments of the H1N1pdm09 viruses. In this sense the material is suitable for publication in eLife.The hypothesis is that viruses able to be spread by an airborne route have hemagglutinin glycoproteins (HA) that undergo the low pH-induced conformational change of the HA required for fusion of the virus envelope with the membranes of the cell at a reduced pH. At first examination this was not apparent from the simple phenotypes of the viruses chosen for study but a detailed examination of the viruses supports this hypothesis.1) The results compared the low pH induced show that human H1N1pdm09 viruses and swine viruses with the HA derived from the human virus displayed a lower pH of fusion of infected vero cells than did the swine viruses chosen to exemplify the swine gamma lineage viruses. The authors then chose examples of swine viruses to examine in more detail, two pairs being chosen that showed differences in their pH of fusion within the pair.Receptor binding, fusion and replication properties of these viruses were established and then examined in transmission studies between infected ferrets, direct contact ferrets and ferrets infected by an airborne route.

This comment describes the results and makes no suggestions.

2) At first examination of the results seemed not to be consistent with the hypothesis. What was seen was that in Pair 1 seemingly more efficient transmission in the virus showing a reduced pH of fusion was not apparent from the simple phenotypes of the viruses chosen for study. The results (Figure 5) show from pair 1 virus G15 showed better or earlier infection of the ferrets by an airborne route than did virus P4, notably as assessed by serology. Whether these results are statistically significant differences need to be established. In Figure 7 the results of the ferret transmission from the other pair of viruses is shown. Here virus P24, with the lower pH of fusion, showed now evidence for transmission in ferrets by the airborne route whilst P19 resulted in transmission to one of the three ferrets exposed by the airborne route. No statistical analysis were presented. In both cases the virus with the higher pH of fusion was the one that seemed to transmit by the airborne route more efficiently – taken at face value.

We added the following sentence to the Figure 5 legend, “The difference of 3/3 versus 1/3 airborne-transmission events was not statistically significant (P > 0.05).”. We added the following sentence to the Figure 7 legend, “The difference of 1/3 versus 0/3 airborne-transmission events was not statistically significant (P > 0.05).” P values for the titers and peak titer days are included in the Results section and in Table 1.

3) However, more work was done to examine the properties of the virus that was able to be transmitted by the airborne route. This was done for virus recovered from the P19/P24 pair. Here the virus collected from donor ferrets, contact ferrets and airborne infected ferrets were examined. The results shown in Figure 8 seemed to show that virus form the contact-infected and the airborne-infected ferrets from the donor infected with P19 showed a lower pH of fusion than seen in some of the donor ferrets whilst this drop in pH of fusion was not seen in the directly contact ferrets infected from the P24 infected ferrets. The results shown in Figure 8C are consistent with the proposal that transmission might be linked to a polymorphism at residue 117 of HA2. However, selection of N117 of HA2 was observed in the donor ferrets. The selection of the variant was therefore not clear cut.

We added the following explanation to the Results section: “HA activation pH values for 2/3 of the P19 donor ferrets remained at pH 5.9 despite containing a majority of the stabilizing HA2-N117 variant. High-pH syncytia formation for these samples was most likely induced by the remaining minority population of the HA2-T117 variant.”

4) Overall, the hypothesis that a low pH of fusion promotes virus transmission is supported by the work described when examined in detail. The hypothesis is not supported without the very detailed examination of the properties of the recovered material from the infected ferrets in one of the two experimental transmissions. However, no further testing of the authors' hypothesis has been done on the recovered viruses to examine their transmission between ferrets.

We added the following paragraph to the Discussion and plan to do the experiment in the future when we are able:

“Additional supporting data from follow-up studies could support the conclusion of this study that HA stabilization is necessary but not sufficient for airborne transmission of swine gamma viruses between ferrets. Samples obtained from nasal washes in this experiment could be plaque purified to isolate G15-HA1-N210, G15-HA1-S210, and P4-HA1-S210. G15-HA1-N210 and G15-HA1-S210 are identical except for the stability-altering residue at HA1-210; therefore, differences in replication and transmission in ferrets between the two viruses would map exclusively to the HA protein. Comparing infection with G15-HA1-S210 and P4-HA1-S210, which have identical HA genes but differ at polymerase residues PA-271 and PB2-648, may reveal the importance of the polymerase genes in supporting airborne transmission of the related swine gamma viruses.”